# Nutation Excitations in the Gyrotropic Vortex Dynamics in a Circular Magnetic Nanodot

**DOI:** 10.3390/nano13030461

**Published:** 2023-01-23

**Authors:** Zukhra Gareeva, Konstantin Guslienko

**Affiliations:** 1Institute of Molecule and Crystal Physics, Subdivision of the Ufa Federal Research Centre of the Russian Academy of Sciences, 450075 Ufa, Russia; 2Institute of Physics and Technology, Bashkir State University, ul. Z. Validi 32, 450076 Ufa, Russia; 3Departamento de Polímeros y Materiales Avanzados: Física, Química y Tecnología, Universidad del País Vasco, UPV/EHU, 20018 San Sebastián, Spain; 4Euskal Herriko Unibertsitatea Quantum Center, University of the Basque Country, UPV/EHU, 48940 Leioa, Spain; 5IKERBASQUE, the Basque Foundation for Science, 48009 Bilbao, Spain

**Keywords:** soft magnetic materials, nanodots, magnetic vortex, gyrotropic dynamics, nutations

## Abstract

A significant activity is devoted to the investigation of the ultrafast spin dynamic processes, holding a great potential for science and applications. However, a challenge of the understanding of the mechanisms of underlying spin dynamics in nanomaterials at pico- and femtosecond timescales remains under discussion. In this article, we explore the gyrotropic vortex dynamics in a circular soft magnetic nanodot, highlighting the impacts given by nutations in the high-frequency part of the dot spin excitation spectrum. Using a modified Thiele equation of the vortex core motion with a nutation term, we analyze the dynamic response of the vortex to an oscillating magnetic field applied in the dot plane. It is found that nutations affect the trajectory of the vortex core. Namely, we show that the directions of the vortex core motion in the low-frequency gyrotropic mode and the high-frequency nutation mode are opposite. The resonant frequencies of gyrotropic and nutational vortex core motions reveal themselves on different scales: gigahertz for the gyrotropic motion and terahertz for the nutations. We argue that the nutations induce a dynamic vortex mass, present estimates of the nutational mass, and conduct comparison with the mass appearing due to moving vortex interactions with spin waves and Doering domain wall mass.

## 1. Introduction

Quite recently, the topic of nutations in the magnetization dynamics have attracted essential attention due to the resonance effects that manifest themselves in the high-frequency range at pico- and femtosecond timescales [1,2,3,4,5,6,7,8]. Ultrafast spin dynamics remains the subject of crucial importance for the spintronic applications and high-speed spintronic devices based on magnetic nanomaterials. Understanding the mechanisms of high-frequency magnetization oscillation processes in soft magnetic nanodots contributes to this field.

The phenomenological Landau–Lifshitz equation of magnetization motion becomes invalid at the short, pico-second and sub-pico-second time scales. The origin of that is the importance of other fast degrees of freedom (electrons, phonons, etc.) on such short time scales, which start to strongly interact with the moving magnetization. To account for other degrees of freedom in magnetic materials and extend the applicability of the Landau–Lifshitz equation to ps and sub-ps spin dynamics, an extra “inertial” term containing the second derivative of the magnetization with respect to time was added. This new term is responsible, in particular, for the magnetization of high frequency THz motions (nutations). Researchers, inspired by the ideas presented in Refs. [9,10], considered several aspects of magnetization dynamics in the inertial mode, including spin waves, ferromagnetic resonance, *s–d* electron interactions, spin current torques, etc. [2,7,11,12,13,14,15]. As it appears, the nutations can be treated by an effective mechanism, allowing them to explain the features of ultrafast magnetization dynamics related, in particular, to the impact of magnetization “inertia”. It was assumed that the nutation effects can be described on the base of a term proportional to the 2nd order derivative of the magnetization, with respect to time, multiplied by the coefficient, η [9]. In most publications, η was considered to be proportional to the magnetization dissipation parameter (α) and nutation time (τ) [4,7,13,16,17], in several others, η is taken as an α-independent parameter [8,18]. The emergence of the nutation term can be interpreted on the basis of the angular momentum conservation law [17] and the magnetization precession in the presence of neighboring magnetic atoms in a crystalline environment. If the system is affected by an external magnetic field, or induced, for example, by the electric current, then this external field tilts the angular momentum axis; in other words, it induces the precession while magnetic moments linked with the neighbors in the crystal environment react inertially and begin to nutate on a femtosecond time scale because they are coupled by the Heisenberg exchange interaction. The nutation is considered to be important if the time scale of the magnetic field change is smaller than the angular momentum relaxation time. Various mechanisms responsible for the emergence of a nutation term were considered in [3,9,12,15,17,19,20], where it was shown that, along with inertia nutations, the first-mentioned Josephson nutation, Rabi oscillations, surface magnetic anisotropy, and geometrical confinement can be reasoned. Approximate estimates of the nutation times (τ) performed in [9,16] result in values of about 10–100 fs. However, in [1], essentially higher essential values of τ, collected experimentally and approximately in the order of 10 ps were extracted from the resonance linewidth. The value of τ estimated from the time-resolved magneto-optical measurements of Co-thin films in [5] is about of 1 ps.

Despite a fairly significant amount of theoretical work considering the nutation in magnetic materials, the experimental observation of the effect encounters difficulties related with the separation of the dynamic responses of the system for different timescales. For example, the frequency difference between the frequencies of ferromagnetic resonance (4 GHz) and nutation resonance (0.6 or 0.4 THz) in CoFeB or permalloy-thin magnetic films is about 100 times [1]. These first experiments on magnetization nutation observed at the frequencies close to 0.5 THz in CoFeB and Ni_81_Fe_19_ films deposited on different substrates were reported very recently [1]. This stimulated further experimental research in this field. Thus, high-order nutation resonances, the dependence of nutation effects on magnetic anisotropy, and the ballistic switching of magnetization have been experimentally detected [5,6]. We note that only uniformly magnetized magnetic samples were considered before. In this regard, it is of interest to explore the influence of nutation on the motion of swirling topologically non-trivial magnetization textures on the nanoscale: magnetic vortices and skyrmions. Prerequisites for such studies can be found in the work in [3,4,17], with estimations of the nutation timescales and lifetimes in magnetic nanostructures [17], nutation interpretation [21] of two excitation modes in the spectrum of the coupled magnetic vortices in magnetic spherical shells [21] and tri-layer magnetic nanodots [22], resonance peaks attributed to precessional and nutation motion in the nanoparticles [3,23], and nutation wave considered as a result of collective spin excitations in the spatially extended spin systems [4].

The vortex gyrotropic mode and its frequency in thin soft magnetic nanodots are well understood, see [24] and the review in [25]. However, when referring to nutations in the magnetic vortex dynamics, one should keep in mind that the inertia term, which is important for nutations, is also associated with the mass of magnetic texture revealed in the spectrum of oscillations in nanosized particles. Thus, as was shown in [26], the lowest modes in the vortex excitation spectra in circular soft magnetic nanodots are owed to the effective vortex mass appearing due to spin waves–vortex interactions.

In order to clarify the differences in the inertially induced oscillations, in this article, we explore the nutation effect on the gyrotropic vortex dynamics. We consider the excitations of the vortex ground magnetic state in a cylindrical soft magnetic nanodot with radius *R* and thickness *L*, calculate the gyrotropic and nutation mode frequencies and the mode intensities, and evaluate their behavior depending on the system parameters. We also show that the nutation term can be treated as dynamic vortex mass in the particle-like equation of the motion of the vortex core.

## 2. Materials and Methods

In this section, we analyze the features of the inertial gyrotropic dynamics of a magnetic vortex in a soft magnetic (permalloy, FeNi alloy) nanodot using the methods of collective variables and determine the main parameters of vortex gyrotropic and nutation resonances arising in the low-frequency (gigahertz) and high-frequency (terahertz) ranges.

Let us consider a cylindrical ferromagnetic nanodot with thickness *L* order of 10 nm and radius *R* order of 100 nm (Figure 1). To explore the magnetic vortex dynamics, we appeal to the Thiele equation of the vortex center motion with a nutation term derived from the corresponding Landau–Lifshitz–Gilbert (LLG) equation:(1)$$\frac{dM}{dt}=-\left|\gamma \right|\left[M\times H_{eff}\right]+\frac{\alpha }{M_{s}}\left[M\times \frac{dM}{dt}\right]+\frac{\eta }{M_{s}}\left[M\times \frac{d^{2}M}{dt^{2}}\right],$$
where $H_{eff}=-\delta E/\delta M$ is the effective magnetic field, and
$$\;\;\;\;\;\;\;\;\;\;E\left(m\right)=A{\left(\partial_{\mu }m_{\alpha }\right)}^{2}-\frac{1}{2}M_{s}m\cdot H_{m}-M_{s}m\cdot H$$
is the magnetic energy density; *γ* is the gyromagnetic ratio; *α* is the Gilbert dissipation parameter; *η* is the nutation coefficient; *M_s_* is the saturation magnetization; *A* is the isotropic exchange stiffness constant proportional to the Heisenberg exchange integral; *α, μ =x, y, z*, $m=M/M_{s}$ is the unit magnetization vector; **H**_m_ is the magnetostatic field; **H** is external magnetic field. No magnetic anisotropy is included to the energy density *E*, assuming a soft magnetic material. Further we consider in-plane circularly polarized oscillating field **H**. We assume that the dot is thin enough and the magnetization ***M***(***ρ****, z, t*) does not depend on the thickness coordinate, *z*. We address to the collective variable approach with magnetization taken in a form ***M***(***ρ****,t*) = ***M***(***ρ****, **X***(*t*)), where ***ρ*** is the radius vector and ***X*** is the vortex core position in the dot plane, respectively.

We use the angular parameterization for the dot magnetization components via spherical angles Θ,Φ: $m_{z}=cos\Theta$ and $m_{x}+im_{y}=sin\Theta exp\left(i\Phi \right)$. After several transformations using the ansatz ***m***(***ρ****,t*) = ***m***(***ρ****, **X***(*t*)), the LLG Equation (1) acquires the form of the modified Thiele equation:(2)$$G\times V-\nabla_{X}W+\hat{D}\cdot V-\hat{N}\frac{dV}{dt}+\mu \left[z\times H\right]=0$$
with the coefficients determined as follows, V=dX/dt, $W\left(X\right)=\int d^{3}rE\left(m\left(r,X\right)\right)$, and the energy density *E* taken here without the Zeeman energy.

The gyrocoupling tensor (emergent magnetic field tensor) is:$$G_{ij}=\frac{M_{s}L}{\left|\gamma \right|}\int d^{2}\rho m\cdot \left[\partial_{i}m\times \partial_{j}m\right]=\frac{M_{s}L}{\left|\gamma \right|}\int d^{2}\rho \sin\Theta \left[\partial_{i}\Theta \partial_{j}\Phi -\partial_{i}\Phi \partial_{j}\Theta \right]$$

The damping tensor and nutation tensor are defined as:$$D_{ij}=-\alpha \frac{M_{s}L}{\left|\gamma \right|}\int d^{2}\rho \left[\partial_{i}m\cdot \partial_{j}m\right]=-\alpha \frac{M_{s}L}{\left|\gamma \right|}\int d^{2}\rho \left[\partial_{i}\Theta \partial_{j}\Theta +sin^{2}\partial_{i}\Phi \partial_{j}\Phi \right],$$
$$N_{ij}=\eta \frac{M_{s}L}{\left|\gamma \right|}\int d^{2}\rho \left[\partial_{i}m\cdot \partial_{j}m\right]=\eta \frac{M_{s}L}{\left|\gamma \right|}\int d^{2}\rho \left[\partial_{i}\Theta \partial_{j}\Theta +sin^{2}\partial_{i}\Phi \partial_{j}\Phi \right],$$
where $\partial_{i}=\partial /\partial X_{i}$ are derivatives with respect to the vortex core **X** position components, and integration is conducted over the dot plane.

Equation (2) is the generalized Thiele equation [27] with the additional nutation term $\hat{N}\frac{dV}{dt}.$ Important difference from the original Thiele equation obtained for the steady state rigid domain wall center motion is that the gyrocoupling, damping, and nutation tensors are calculated via derivatives, with respect to the soliton center position **X**, not with respect to the current coordinate, r=(x,y,z). The gyrovector is determined as a dual vector for the antisymmetric tensor $G_{ij}$; $G=–Gz,\;G=G_{xy}=-G_{yx}=\frac{2\pi }{\left|\gamma \right|}qpLM_{s}$ where the integer number *q* is vorticity, *p* = ±1 is the vortex polarization that determines the direction of *m_z_* component in the center of vortex (here, we take *q* = +1, *p* = +1, *G* > 0, as in [28,29]), ***z*** is the unit vector directed along *z*-axis, parallel to the dot thickness. The second and third terms in Equation (2) are the restoring and dissipative forces. Following the approach elaborated in [24,28,29], we extract the Zeeman force as the separate term $F_{H}=\;\mu \left[z\times H\right]$, where the coefficient *μ* calculated within the two-vortex model is $\mu =\pi RL\xi CM_{s}$ and ξ = 2/3 [24], and *C* = ±1 determines vortex chirality.

For the vortex in-plane circular motion, excited by the in-plane circularly polarized oscillating magnetic field $H=\left(H_{x},H_{y}\right)$, $H\left(t\right)=H_{x}+iH_{y}=H_{0}exp\left(i\omega t\right)$, the vortex velocity and acceleration are expressed via the angular velocity **ω** = ω***z***, $V=\frac{dX}{dt}=\left[\omega \times X\right]=\omega \left[z\times X\right]$, $\frac{dV}{dt}=-\omega^{2}X$ that allows us to rewrite Equation (2) in terms of the oscillation frequency ω:(3)$$-\hat{N}\omega^{2}X-G\times \left[\omega \times X\right]+\nabla_{X}W-\hat{D}\cdot \left[\omega \times X\right]-\mu \left[z\times H\right]=0$$

In the case of sufficiently small displacements of the vortex core from its equilibrium position ***X*** = (*X, Y*), the magnetic energy can be decomposed in the series of small parameter as $W\left(X\right)=W\left(0\right)+\kappa \left(X^{2}+Y^{2}\right)/2$, where κ is the stiffness coefficient determined for the vortex magnetic state in [29] within an appropriate displaced vortex core model.

It was shown in [24] that for cylindrical nanodot, the damping and nutation tensors are diagonal at **X** = 0 with equal diagonal components. We use notations $D_{xx}=D_{yy}=D$ and $N_{xx}=N_{yy}=N$. We note that within the rigid vortex model ***m***(***ρ****, **X***(*t*)) = ***m***(***ρ***
*− **X***(*t*)); $\partial /\partial X_{i}=-\partial /\partial x_{i};$ and, therefore, $Tr\left(\hat{D}\right)$, $Tr\left(\hat{N}\right)$ are proportional to the dot exchange energy. Then, Equation (3) written in terms of the vortex position ***X*** reads as:(4)$$-N\omega^{2}X-G\omega X+\kappa X-D\left[\omega \times X\right]-\mu \left[z\times H\right]=0$$
or in terms of the vector components:(5)$$X\left(-\frac{N}{G}\omega^{2}-\omega +\frac{\kappa }{G}\right)+\frac{D}{G}\omega Y=-\frac{\mu }{G}H_{y},-\frac{D}{G}\omega X+Y\left(-\frac{N}{G}\omega^{2}-\omega +\frac{\kappa }{G}\right)=\frac{\mu }{G}H_{x}.$$

It is convenient to rewrite Equation (5) using the complex variables for the vortex core position and oscillating magnetic field, Z=X+iY, $H=H_{x}+iH_{y},$, as:(6)$$Z=i\frac{\mu }{G}\frac{H}{\left(\omega_{0}-\omega -\eta_{G}\omega^{2\;}+id\omega \right)},$$
where $\omega_{0}=\kappa /G,\;\eta_{G}=N/G,\;d=-D/G$.

Introducing the circular component $\langle M_{+}\rangle =M_{x}+iM_{y}$ of the volume averaged dot magnetization 〈M(t)〉=−(μ/V)[z×X(t)], we get the simple relation $\langle M_{+}\rangle =-i\left(\mu /V\right)Z$. Then, substituting Solution (6) into this relation and introducing the Fourier transforms $\langle M_{+}\left(t\right)\rangle =\langle M_{+}\left(\omega \right)\rangle exp\left(i\omega t\right),\;H\left(t\right)=H\left(\omega \right)exp\left(i\omega t\right),$ we obtain the linear relation:(7)$$\langle M_{+}\left(\omega \right)\rangle =\frac{\mu^{2}}{GV}\frac{H\left(\omega \right)}{\left(\omega_{0}-\omega -\eta_{G}\omega^{2}+id\omega \right)}=\chi \left(\omega \right)H\left(\omega \right)$$
Allowing us to obtain the dynamic circular magnetic susceptibility χ(ω) in the form:(8)$$\chi \left(\omega \right)=\chi \prime \left(\omega \right)+i\chi \prime \prime \left(\omega \right)=\frac{\mu^{2}}{GV}\frac{\omega_{0}-\omega -\eta_{G}\omega^{2}-id\omega }{{\left(\omega_{0}-\omega -\eta_{G}\omega^{2}\right)}^{2}+d^{2}\omega^{2}}$$

The imaginary part of χ(ω):(9)$$\chi \prime \prime \left(\omega \right)=-\chi \left(0\right)\frac{d\omega \omega_{0}}{{\left(\omega_{0}-\omega -\eta_{G}\omega^{2}\right)}^{2}+d^{2}\omega^{2}},$$
determines the intensities and linewidths of magnetic resonances. Here, $\chi \left(0\right)=\mu^{2}/VG\omega_{0}$ is the static vortex dot susceptibility calculated in [24,29].

The eigenfrequencies of the vortex motion are determined by the poles of dynamical susceptibility given by Equation (7):(10)$$\eta_{G}\omega^{2}+\left(1-id\right)\omega -\omega_{0}=0.$$

The quadratic Equation (10) has complex roots $\omega_{1,2}$, whose real parts determine the frequencies of the gyrotropic vortex motion *ω_G_* and vortex nutation *ω_η_,* and the imaginary parts determine damping (linewidths) of these excitation modes. We search for the solutions of Equation (10) in the form ω=ω′+iω″ and find assuming ω″<ω′, the resonance frequencies to be $\omega_{res}=\omega \prime \left[1+id/\left(1+2\eta_{G}\omega \prime \right)\right]$, where ω′ satisfies the equation $\eta_{G}{\omega \prime }^{2}+\omega \prime -\omega_{0}=0$. In the limit $\eta_{G}\omega_{0}\;$<< 1, this equation has two solutions: $\omega_{G}=\omega_{0}$, standing for the vortex gyrofrequency of the counter-clockwise vortex core motion, and $\omega_{\eta }=-\frac{1}{\eta_{G}}$, which is responsible for the nutation frequency of the clockwise core motion. The corresponding complex roots of Equation (10) are $\omega_{1}=\omega_{0}\left(1+id\right)$ and $\omega_{2}=-\left(1/\eta_{G}\right)\left(1-id\right)$. The real parts of the roots, $\omega_{0}>0$ and $\omega_{\eta }<0$, have opposite signs due the positive sign of the nutation parameter $\eta_{G}=N/G$ (N>0  due to η>0 [9,12] and G>0).

The low frequency vortex core precession and high-frequency nutation motions are in the opposite directions. More definitely, for a magnetic vortex with positive polarization (p>0), the gyrotropic motion of the vortex core is counter-clockwise, while the nutation motion is clockwise. For a vortex with negative polarization (p<0), the situation is reversed.

## 3. Results and Discussion

Obtained analytical expressions for the dynamical susceptibility χ(ω) of the vortex-state nanodots (8,9) are similar to those calculated [2,7] or simulated [12] recently for uniformly magnetized samples. However, as noticed in [24], all parameters in the dynamical susceptibility of the vortex state dot (intensities, resonance frequencies, linewidths) are essentially different from ones for uniformly magnetized particles. The developed model of the magnetic susceptibility is valid for all parameters of the vortex nanodot ($G,\;D,\;N,\;\omega_{0},\;\mu$). The gyrovector *G* does not depend in the linear approximation on the particular model of the displaced vortex core. However, the damping *D* and nutation *N* coefficients, the vortex gyrotropic frequency $\omega_{0\;},$ and the coefficient *μ* of response to magnetic field are model dependent. The dependence of *D* and *N* on the particular model of the displaced vortex is relatively weak and can be neglected in the first approximation. The model dependence of $\omega_{0\;}$ and *μ* is strong. For all numerical calculations (plots) and estimations, we used the two-vortex model of the displaced vortex described in detail in [24,25].

Here, we calculate numerically the dependences of *ω_G_*/2π (GHz) and *ω_η_*/2π (THz) on the nutation parameter *η_G_*, as shown in Figure 2, for the permalloy (FeNi alloy) cylindrical nanodot with the magnetic material parameters *A* = 1.1·10^−6^ erg/cm, *M_s_* = 800 G, *α* = 0.01, γ/2π = 2.95 MHz/Oe, dot radius *R* = 100 nm, and dot thickness *L* = 10 nm, as in [25,29]. The damping parameter is $D=-2\pi \alpha M_{s}L\left(5/8+\ln\left(R/R_{c}\right)/2\right)/\gamma$ [30], the vortex core radius is $R_{c}=0.68L_{e}{\left(L/L_{e}\right)}^{1/3}$, where $L_{e}=\sqrt{2A}/M_{s}$ is the exchange length. Therefore, according to [29] for the nanodot aspect ratio *L/R* = 0.1 we take $\omega_{0}/2\pi =$ 0.52 GHz, and *d* = 0.02 (*R_c_* = 18.5 nm). The range of the nutation parameter *η_G_* is related with sub-picosecond timescales [9,16,17]. Direct calculations within two vortex model yields the relation $\eta_{G}=\left(5/8+\ln\left(R/R_{c}\right)/2\right)\eta$ or $\eta_{G}\approx 2\eta$ for our nanodot parameters. The estimation of the nutation vortex frequency for these dot parameters and *τ* = 10 ps yields is *ω_η_*/2π = 0.9 THz.

The vortex gyrotropic (*ω_G_*) and nutation (*ω_η_*) eigenfrequencies presented in Figure 2 are essentially different. Therefore, for convenience, we use a logarithmic frequency scale. Note that we are considering the absolute values of *ω*, since the sign of the frequency, as noted above, is important only for determining the sense of the vortex core rotation. As shown in Figure 2, the gyrotropic frequency *ω_G_* of the order of 1 GHz very slowly decreases with the increase in the nutation parameter *η_G_*. The absolute value of the nutation frequency *ω_η_* changing in the THz range decreases. If we choose the vortex core polarization *p* = +1, the low-frequency vortex core gyrotropic motion is counter-clockwise (positive resonance frequency), whereas the high-frequency nutation motion is clockwise (negative resonance frequency). The resulting core motion is a superposition of two such rotations. The vortex core trajectory for the vortex gyroscopic and nutation modes can be represented as $Z_{G}\left(t\right)=Z_{1}exp\left(i\omega_{0}t\right)exp\left(-d\omega_{0}t\right)$ and $Z_{\eta }\left(t\right)=Z_{2}exp\left(-it/\eta_{G}\right)exp\left(-dt/\eta_{G}\right)$, respectively. Therefore, to excite and detect the gyrotropic and nutation spin modes in the vortex, such as structures, circularly polarized oscillating magnetic fields of different directions are needed.

Note that the differences in the senses of rotation of two spin modes, where the first one is due to spin precession, and the second one is due to the spin inertia, is a common feature of the spin dynamics in ferromagnets. The first mode corresponds to the ferromagnetic resonance in the uniformly magnetized samples [2] or to the gyrotropic resonance in curling magnetic structures [29]. The second high-frequency mode emerges due to the spin inertia and is manifested as the nutation spin motion. The resulting spin dynamics is a superposition of precession and nutation. In both cases, ferromagnetic resonance [2] and vortex gyrotropic motion, the magnetization oscillates in the plane *xOy* perpendicularly to the selected axis *Oz*, which is dictated by a strong *dc* magnetic field or by the vortex core polarization direction, respectively. In the first case, the oscillation magnetization is spatially unform [2], whereas in the second case it is inhomogeneous and volume-averaged magnetization 〈M(t)〉, defined in the text below Equation (6) [24]. The opposite signs of the gyrotropic and nutation resonance frequencies result in the different signs of magnetic susceptibilities shown in Figure 3. The dynamical susceptibility χ(ω)  defined by Equation (8) is marked as $\chi_{+}\left(\omega \right)$ in [2]. The nutation resonance frequency in [2] corresponds to a peak at a positive frequency ω>0  of the susceptibility $\chi_{-}\left(\omega \right)$, which is in response to the magnetic field circulating in the opposite (clockwise) direction. Both approaches are equivalent because of the relation $\chi_{-}\left(\omega \right)=\chi_{+}\left(-\omega \right)$.

The nutation relaxation rate is $d/\eta_{G}=\alpha /\eta$. If we use the definition of the nutation parameter *η* via the nutation time τ as η=ατ, then the nutation mode trajectory is $Z_{\eta }\left(t\right)=Z_{2}exp\left(-it/\eta_{G}\right)exp\left(-t/\tau \right)$. The nutation vortex motion decays with the characteristic time τ, i.e., very fast, on the picosecond time scale. The corresponding resonance linewidth 1/τ is high, in the order of 0.5–1 THz. Such large linewidths were experimentally detected for the nutation resonance in the CoFeB and FeNi thin films in [1].

Equation (9) for the imaginary part of the vortex dynamic susceptibility allows us to calculate the ratio of the peak intensities at the nutation and gyrotropic resonance as $\chi \prime \prime \left(\omega_{\eta }\right)/\chi \prime \prime \left(\omega_{0}\right)=\omega_{0}/\omega_{\eta }$. The nutation frequency $\omega_{\eta }=\eta^{-1}{\left[5/8+\ln\left(R/R_{c}\right)/2\right]}^{-1}$ is much larger (approximately in 10^3^ times) than the vortex gyrotropic frequency $\omega_{0}$. Therefore, the high-frequency nutation resonance peak value is very small in comparison with the low-frequency vortex gyration peak.

The dependence of χ″ on the frequency for the value of the nutation parameter selected in accordance with the experiments *η_G_* = 0.1 ps is shown in Figure 3; in order to highlight the nutation peak of the susceptibility at high frequencies, we inserted it into the inset to Figure 3. Note that only the $\chi_{\max}^{\left(1\right)}=\chi \prime \prime \left(\omega_{G}\right)$ peak exists in the frequency dependence of the imaginary part of dynamical susceptibility χ″(ω) when nutation is supposed to be absent. An additional nutation peak $\chi_{\max}^{\left(2\right)}=\chi \prime \prime \left(\omega_{\eta }\right)$ emerges when the nutation parameter *η_G_* acquires non-zero values, it tends to shift to the low-frequency part of the spectrum when increasing the nutation parameter *η_G_*.

In several published papers, it is stated that the nutation is equivalent to some of the inertia of the spin system. The nutation moment of inertia was introduced in [9,10,11] and then it was calculated in [14] and estimated in [11,15], However, the corresponding inertial mass was never calculated in ferromagnets, except in the specific case of interaction of moving magnetic vortex with a bath of spin waves [26,31] or spin *sd*–exchange interaction with conductivity electrons in ferromagnetic metals [15]. The nutation mass introduced in [4] for spin waves has a different physical sense. It is just the nutation spin wave frequency at zero wave vector. The inertia of the spin in metallic ferromagnets represents a nonadiabatic contribution from the environmental degrees of freedom (conductivity electrons). It was shown in [10,26] that such external degrees can be spin waves excited over inhomogeneous magnetization (vortex) background. The origin of the inertial mass in both cases is dynamical. From the other side, an inertial mass was introduced to magnetism by Doering in 1948 considering a rigid motion of a 1D domain wall in anisotropic ferromagnet [32] without any other degrees of freedom.

Here, we consider the connection between the nutation term in the LLG equation of motion (1) and the inertial vortex (2D topological magnetic soliton) mass. The nutation term defined by the second order time derivative of magnetization contributes to a vortex mass determined by the nutation coefficient *N* in the modified Thiele equation for the vortex center position X motion. Equations (2)–(4) allow us to assume that the nutation coefficient *N* can be treated as a vortex inertial mass in the course of particle-like motion of the vortex core, described by the core position **X**, velocity *d***X**/*dt*, and acceleration d^2^**X**/dt^2^. Then, the vortex mass can be explicitly written as:(11)$$M_{\upsilon }=\frac{1}{2}\frac{L}{\gamma^{2}}\alpha \omega_{M}\tau \left[\frac{5}{8}+\frac{1}{2}ln\left(\frac{R}{R_{c}}\right)\right],$$
where $\omega_{M}=4\pi \gamma M_{s}$.

It can be seen from this expression that nutation-induced vortex mass is directly proportional to the nutation time *τ* and depends on the sizes of nanodot and the parameters of the magnetic material. Using Equation (11), we can estimate the vortex mass for a definite system. The range of vortex mass of permalloy nanodot (*R* = 100 nm, *L* = 10 nm) related with the range of nutation parameter *η* = *ατ*~0.01 × ($10^{-12}-10^{-11}$ s) is $10^{-23}-10^{-22}$ g. Although this nutation vortex mass is essentially smaller than the dynamical vortex mass emerging due to the spin wave–vortex interaction considered in [10,26], it has the same order of magnitude as the Doering mass of domain walls in uniaxial anisotropic ferromagnets. We can estimate the Doering mass for the vortex in soft magnetic material using its definition $M_{D}=S/\left(2\pi \gamma^{2}\Delta \right)$, where S≈2RL is the domain wall square and Δ≈R is the domain wall width, and we can obtain the value $M_{D}=L/\left(\pi \gamma^{2}\right)$ comparable with the vortex nutation mass given by Equation (11).

## 4. Conclusions

We considered the high-frequency vortex excitations in the framework of the nutation approach to the inertial dynamics in ferromagnets. Based on the Thiele equation with a nutation term obtained here and the model of vortex dynamics developed in [24], we calculated the resonant frequencies and the dynamic susceptibility of a single vortex state in a magnetically soft nanodot to excitations induced by an in-plane oscillating magnetic field with a circular polarization. We showed the existence of two resonant frequencies reflected in two peaks of the dynamic magnetic susceptibility, corresponding to the vortex core gyrotropic motion and the nutations. The first frequency is the standard gyrotropic frequency, the value of which belongs to the gigahertz range; the second frequency is responsible for spin nutations manifested themselves in the terahertz frequency range. The magnitude of the nutation frequency and the intensity of the dynamic response depend on the timescales of the nutation effects; the longer the nutation time, the lower the nutation frequency and the more pronounced the response signal.

A distinctive feature of the considered vortex oscillations is the difference in the directions of spin rotation. Vortex core in the gyrotropic and nutation modes rotates in opposite directions. Namely, for a vortex with positive polarity, the low frequency gyrotropic motion is counterclockwise, while the high-frequency nutation motion is clockwise; for a vortex with negative polarity, the situation is reversed. The resulting trajectory of the vortex core represents superposition of precessional and nutational motion. Therefore, the vortex trajectory in a nanodot is affected by the nutations. As follows from our calculations for the detection and excitation of the low-frequency and high-frequency modes of the vortex state in the circular thin nanodot, it is necessary to apply circularly polarized oscillating magnetic fields of different directions due to the differences in the sense of magnetization rotation.

The nutation term in the initial LLG equation of magnetization motion leads to the appearance of a finite vortex inertial mass in the particle-like equation of motion of the vortex core. Our estimations showed that the magnitude of nutation-induced vortex mass depends on the nutation time, the sizes of nanodot, and the parameters of magnetic material; for the thin permalloy nanodot (*R* = 100 nm, *L* = 10 nm), it attains a magnitude of about $10^{-23}-10^{-22}$ g. Despite the rather small values of the nutation mass, even in comparison with the mass that arises due to the interaction of the spin wave with the vortex, taking into account its effect is important for the inertial dynamics of magnetization. The presence of a non-zero mass at the vortex can lead not only to a change in the vortex motion trajectory but also to nonadiabatic interactions of magnetic moments, comprising curling magnetic state, with the crystalline environment, which, in turn, can induce additional torques, affecting various aspects of magnetization dynamics.

## Figures and Tables

**Figure 1 nanomaterials-13-00461-f001:**
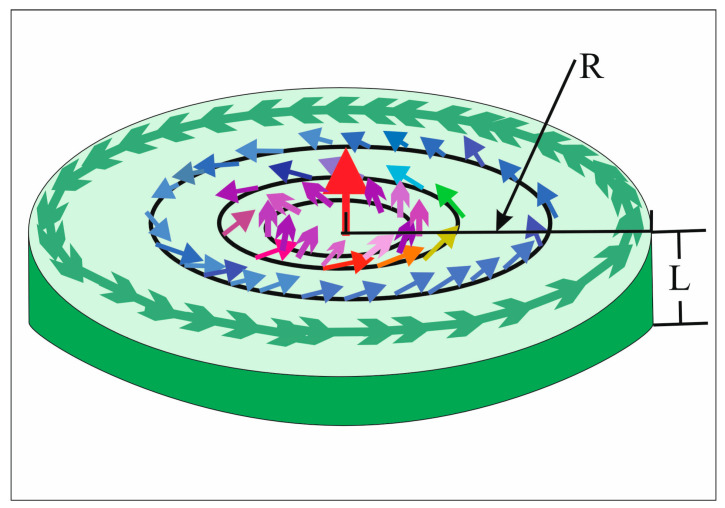
Schematic illustration of the vortex ground state in a circular magnetic nanodot.

**Figure 2 nanomaterials-13-00461-f002:**
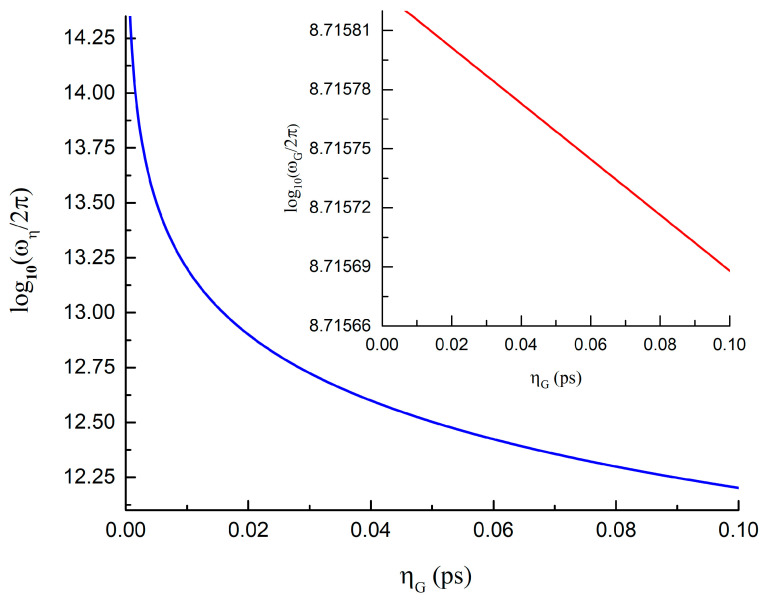
Dependence of the vortex eigenfrequencies *ω_η_* and *ω_G_* (insert), taken in the logarithmic scale on the nutation parameter *η_G_* in permalloy (FeNi alloy) circular nanodot. The nanodot thickness is 10 nm, and the radius is 100 nm.

**Figure 3 nanomaterials-13-00461-f003:**
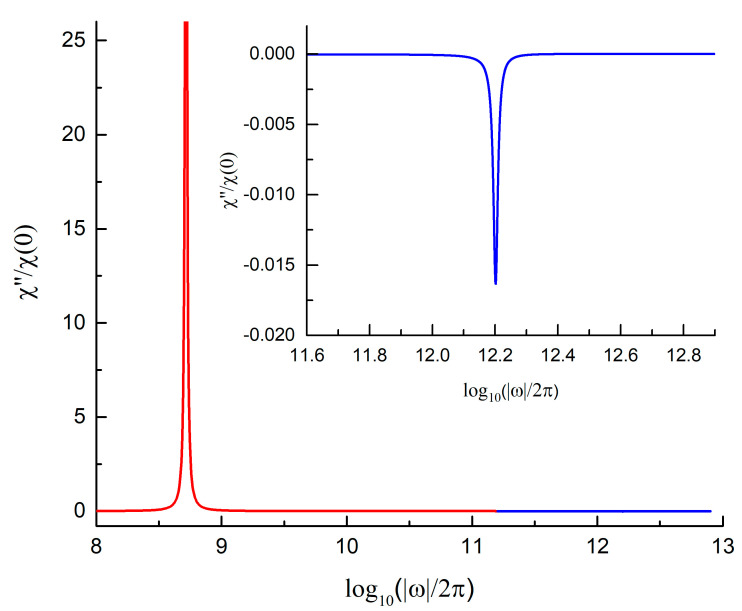
Reduced dynamic susceptibility χ″(ω)/χ″(0) of the vortex state permalloy circular nanodot vs. logarithmic frequency log(|*ω*|/2*π*) plotted at the nutation parameter *η_G_* = 0.1 ps in consistency with the experimental data [1]. The nanodot thickness is 10 nm, and the radius is 100 nm.

## Data Availability

Not applicable.

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
