# Peer review of "Nutation Excitations in the Gyrotropic Vortex Dynamics in a Circular Magnetic Nanodot"

_nanomaterials, 2023, doi:10.3390/nano13030461_

Round 1
Reviewer 1 Report
It is an excellent theoretical paper. The manuscript is clearly written in good scientific English. The topic of the publication, nutation related high frequency gyrotropic vortex dynamics in a circular soft magnetic nanodot, is undoubtedly of considerable interest for the scientific community especially for those specialized in the field of magnetism. Importantly, while the absolute majority of authors concentrate on the influence of the nutation related mechanisms on the conventional spin wave modes, the submitted paper addresses an entirely different type of magnetic dynamics, namely, the high frequency features characterising the nutation motion of magnetic vortices, which makes this theoretical study particularly timely and pertinent. In my opinion, it should be published without delay.
Minor issues…
1. Page 5 line 187 “It convenient to rewrite Equations (5) using…”, probably “It is convenient to rewrite Equations (5) using…”?
2. Page 6 line 237 “displaced vortex is relatively week and…”, probably “relatively weak”?
3. Page 9 line 369 “The authors thank to A. Stashkevitch”, actually, this name is spelt in English as “Stashkevich”.
4. Page 6 bottom paragraph. “If we choose the vortex core polarization p=+1, the low-frequency vortex core gyrotropic motion is counter-clockwise (positive resonance frequency), whereas the high-frequency nutation motion is clock-wise (negative resonance frequency)”. It is a very important finding and I would suggest adding a short paragraph elucidating, in plain terms, the physics behind this important feature for the general reader.

Author Response
We thank the Referee for his/her positive evaluation of our manuscript and the comments which helped us in making the presentation clearer. In response to the referee’s remarks, we have provided the point-to-point responses below.
Reviewer-1 comments:
Remark 1. Page 5 line 187 “It convenient to rewrite Equations (5) using…”, probably “It is convenient to rewrite Equations (5) using…”?
Our reply. We thank to Referee for this remark. We made the suggested correction.
Remark 2. Page 6 line 237 “displaced vortex is relatively week and…”, probably “relatively weak”?
Our reply. We changed the word “week” to “weak”. It is a misprint.
Remark 3. Page 9 line 369 “The authors thank to A. Stashkevitch”, actually, this name is spelt in English as “Stashkevich”.
Our reply. Thank you, we made the requested correction.
Remark 4. Page 6 bottom paragraph. “If we choose the vortex core polarization p=+1, the low-frequency vortex core gyrotropic motion is counter-clockwise (positive resonance frequency), whereas the high-frequency nutation motion is clock-wise (negative resonance frequency)”. It is a very important finding and I would suggest adding a short paragraph elucidating, in plain terms, the physics behind this important feature for the general reader.
Our reply. We thank to Referee for this useful remark. We added a short paragraph on this subject it the end of Section 2 (page 6):
´´ The real parts of the roots, and , have opposite signs due the positive sign of the nutation parameter ( due to [9,12] and ).
The low frequency vortex core precession and high – frequency nutation motions are in the opposite directions. More definitely, for a magnetic vortex with positive polarization ( ), the gyrotropic motion of the vortex core is counter-clockwise, while the nutation motion is clock-wise. For a vortex with negative polarization ( ), the situation is reversed. ´´
and to Sec. 3 (p. 7, after Fig. 2):
” Therefore, to excite and detect the gyrotropic and nutation spin modes in the vortex – like structures, circularly polarized oscillating magnetic fields of different directions are needed.”
“Note that the differences in the senses of rotation of two spin modes, where the first one is due to spin precession, and the second one is due to the spin inertia, is a common feature of the spin dynamics in ferromagnets. The first mode corresponds to the ferromagnetic resonance in the uniformly – magnetized samples [2] or to the gyrotropic resonance in curling magnetic structures [29]. The second high – frequency mode emerges due to the spin inertia and is manifested as the nutation spin motion. The resulting spin dynamics is a superposition of precession and nutation. In both cases, ferromagnetic resonance [2] and vortex gyrotropic motion, the magnetization oscillates in the plane xOy perpendicular to the selected axis Oz, which is dictated by a strong dc magnetic field or by the vortex core polarization direction, respectively. In the first case the oscillation magnetization is spatially unform [2], whereas in the second case it is inhomogeneous and volume-averaged magnetization , defined in the text below Eq. (6), oscillates [29]. The opposite signs of the gyrotropic and nutation resonance frequencies result in the different signs of magnetic susceptibilities shown in Figure 3. The dynamical susceptibility defined by Equation (8) is marked as in Ref. [2]. The nutation resonance frequency in Ref. [2] corresponds to a peak at a positive frequency of the susceptibility , response to the magnetic field circulating in opposite (clock-wise) direction. Both approaches are equivalent because of the relation ”

Reviewer 2 Report
The abstract section lacks an introduction to the research background.
The figures are presented in a quite single way and the fonts in each figure are not uniform.
Author Response
We thank the Referee for his/her positive evaluation of our manuscript and the valuable comments and questions. In the following, we address the comments and suggestions of the Referee. We believe that addressing the comments has improved the manuscript.
Reviewer-2 comments:
Remark 1. The abstract section lacks an introduction to the research background.
Our reply:
Thanks for this comment.
We added a brief background of the research in the abstract. We would like to note that usually abstract is aimed to represent the principal results of a paper and the information on the history and the background of a research is given in Introduction section of the paper.
We added abstract with the sentences: “A significant activity is devoted to investigation of the ultrafast spin dynamic processes, holding a great potential for science and applications. However, a challenge of the understanding the mechanisms of underlying spin dynamics in nanomaterials at pico – and femtosecond timescales remains under discussion. …”
Remark 2. The figures are presented in a quite single way and the fonts in each figure are not uniform.
Our reply: We corrected the figures, also making all fonts to be uniform.

Reviewer 3 Report
In this paper,the authors appeal to the Thiele equation of the vortex center motion,in which the nutation term derived from the Landau – Lifshitz - Gilbert (LLG) equation. Then,the resonant frequency and dynamic susceptibility of a soft nanodot are calculated, the frequency, intensity and vortex chirality of nutaion are also discussed.
This work inherits the previous work and makes some new extensions, gives some new things about nutation in theory. English language and technical terms are fluent and appropriate. But I still have some comments:
1. The authors calculated a simplest model--soft magnetic nanodot,found that the vortex core gyrotropic motion is counter-clockwise, whereas the high-frequency nutation motion is clock-wise. If it is not a symmetrical geometry,are the chirality of vortices always opposite to that of nutations? What are the physical implications of the chirality opposites?
2. In the study of magnetic vortices and skyrmions, magnetic field and current excitation are often used to control the polarity and chirality of the vortices. The evolution processes and rules can be simulated theoretically. Then, if nutation terms are considered in the simulation, will it be helpful to the accuracy of the simulation? Are there some special circumstances in which the simulation results would be different?
3. Page 6, Fig 2 and Page 7, Fig 3: the Ticks on the Axis in or out? That should be in the same format, and the units “ps” should be in parentheses
Author Response
Reviewer 3
We thank the Referee for his/her positive evaluation of our manuscript and the comments which helped us in making the presentation stronger. In response to the referee’s remarks, we have provided the point-to-point responses below.
Remark 1. The authors calculated a simplest model--soft magnetic nanodot,found that the vortex core gyrotropic motion is counter-clockwise, whereas the high-frequency nutation motion is clock-wise. If it is not a symmetrical geometry,are the chirality of vortices always opposite to that of nutations? What are the physical implications of the chirality opposites?
Our reply. We thank Referee for this useful remark. The differences in the handnesses of two spin modes, which for vortices are gyrotropic mode and nutation mode, is the general feature of inertial spin dynamics in ferromagnets. Independent on the symmetrical geometry, the chirality of vortex core motion is always opposite to that of nutations. Due to inertia the directions of the angular momentum and localized spins do not coincide, which can give rise to a plenty of interesting dynamic phenomena. We added to the manuscript two paragraphs: at the end of Section 2 (page 6) and in p. 7 to give more detailed explanation of the phenomena:
´´ The real parts of the roots, and , have opposite signs due the positive sign of the nutation parameter ( due to [9,12] and ).
The low frequency vortex core precession and high – frequency nutation motions are in the opposite directions. More definitely, for a magnetic vortex with positive polarization ( ), the gyrotropic motion of the vortex core is counter-clockwise, while the nutation motion is clock-wise. For a vortex with negative polarization ( ), the situation is reversed. ´´ (p. 6)
” Therefore, to excite and detect the gyrotropic and nutation spin modes in the vortex – like structures, circularly polarized oscillating magnetic fields of different directions are needed.”
“Note that the differences in the senses of rotation of two spin modes, where the first one is due to spin precession, and the second one is due to the spin inertia, is a common feature of the spin dynamics in ferromagnets. The first mode corresponds to the ferromagnetic resonance in the uniformly – magnetized samples [2] or to the gyrotropic resonance in curling magnetic structures [29]. The second high – frequency mode emerges due to the spin inertia and is manifested as the nutation spin motion. The resulting spin dynamics is a superposition of precession and nutation. In both cases, ferromagnetic resonance [2] and vortex gyrotropic motion, the magnetization oscillates in the plane xOy perpendicular to the selected axis Oz, which is dictated by a strong dc magnetic field or by the vortex core polarization direction, respectively. In the first case the oscillation magnetization is spatially unform [2], whereas in the second case it is inhomogeneous and volume-averaged magnetization , defined in the text below Eq. (6), oscillates [29]. The opposite signs of the gyrotropic and nutation resonance frequencies result in the different signs of magnetic susceptibilities shown in Figure 3. The dynamical susceptibility defined by Equation (8) is marked as in Ref. [2]. The nutation resonance frequency in Ref. [2] corresponds to a peak at a positive frequency of the susceptibility , response to the magnetic field circulating in opposite (clock-wise) direction. Both approaches are equivalent because of the relation ” (p. 7)
Remark 2. In the study of magnetic vortices and skyrmions, magnetic field and current excitation are often used to control the polarity and chirality of the vortices. The evolution processes and rules can be simulated theoretically. Then, if nutation terms are considered in the simulation, will it be helpful to the accuracy of the simulation? Are there some special circumstances in which the simulation results would be different?
Our reply. These are interesting questions. However, they are beyond the scope of our manuscript, where we use analytical approach to the problem. The analytical LLG equation with the nutation term was numerically integrated by Ciornei et al., Ref. 9, Olive et al., Ref. 12, 13 (a second order Runge-Kutta algorithm was used). In all cases, the high-frequency nutation resonance peak was found in the uniformly magnetized samples. From the other side, we are not aware any pure numerical solution of this problem using the standard micromagnetic codes like OOMMF or MuMax.
In general, the nutations manifest themselves at high – frequencies in the systems where inertia effects are essential. If one considers such situation, the accounting of nutation term is helpful. Usually, the analytical approach provides benchmarks that should be reproduced in simulations. To have more pronounced effect here, we considered spin excitations induced by the in-plane circularly polarized oscillating magnetic field in magnetic vortex in soft circular nanodot. If one considers another situation the results will be different.
Remark 3. Page 6, Fig 2 and Page 7, Fig 3: the Ticks on the Axis in or out? That should be in the same format, and the units “ps” should be in parentheses.
Our reply. We corrected Figures 2 and 3 making ticks in the same format and put the units “ps” in the parentheses.
